# Effects of rumen-bypass protein supplement on growth performance, hepatic mitochondrial protein complexes, and hepatic immune gene expression of beef steers with divergent residual feed intake

Modoluwamu Idowu[1], Godstime Taiwo[1], Taylor Sidney[1], Emily Treon[1], Yarahy Leal[1], Deborah Ologunagba[1], Francisca Eichie[1], Andres Pech-Cervantes[2], Ibukun M. Ogunade[1] *

1 Division of Animal Science, West Virginia University, Morgantown, West Virginia, United States of America, 2 Division of Agriculture, Food and Resource Sciences, University of Maryland Eastern Shore, Princess Anne, Maryland, United States of America

* Ibukun.ogunade@mail.wvu.edu

## Abstract

We investigated the impact of a rumen-bypass protein (RBP) supplement on growth performance, plasma and urinary N (UN) concentration, hepatic mitochondrial protein complexes, and hepatic mRNA expression of immune genes of beef steers with negative or positive residual feed intake (RFI) phenotype. Forty crossbred beef steers with an average body weight (BW) of 492 ± 36 kg were subjected to a generalized randomized block design over a 42-day experimental period. This study followed a 2 × 2 factorial arrangement of treatments. The factors evaluated were: 1) RFI classification (low-RFI (-2.12 kg/d) vs. high-RFI (2.02 kg/ d), and 2) rumen-bypass protein supplement: RBP supplement (RBP; 227 g/steer/d) vs. control diet (CON; 0 g/d), resulting in four distinct treatments: LRFI-CON (n = 10), LRFI-RBP (n = 10), HRFI-CON (n = 10), and HRFI-RBP (n = 10). The RBP supplement (84% crude protein) is a mixture of hydrolyzed feather meal, porcine blood meal, and DL-methionine hydroxy analogue. The beef steers were stratified by BW, randomly assigned to treatments, and housed in four pens (1 treatment/pen) equipped with two GrowSafe feed bunks each to measure individual dry mater intake (DMI). Body weight was measured every 7 d. Liver tissue samples were collected on d 42 from all the beef steers. These samples were used for mRNA expression analysis of 16 immune-related genes and for evaluating the mitochondrial protein complexes I - V. No significant effects due to RBP supplementation or RFI × RBP interactions (P > 0.05) were observed for average daily gain (ADG) and DMI. However, compared to high-RFI steers, low-RFI steers showed a trend towards reduced DMI (12.9 vs. 13.6 kg/d; P = 0.07) but ADG was similar for the two RFI groups. Regardless of RFI status, supplemental RBP increased blood urea nitrogen (BUN) (P = 0.01), with a lower BUN concentration in low-RFI steers compared to high-RFI ones. A tendency for interaction (P = 0.07) between RFI and RBP was detected for the UN concentrations; feeding the dietary RBP increased the UN concentration in high-RFI beef steers (209 vs. 124 mM),

**Data Availability Statement:** All relevant data are within the manuscript and its Supporting Information files.

**Funding:** This study was funded by West Virginia University Experimental Station in support of U.S. Department of Agriculture hatch multi-state regional project W-3010. The funders had no role in study design, data collection and analysis, decision to publish, or preparation of the manuscript.

**Competing interests:** The authors have declared that no competing interests exist.

whereas the concentration was lower than that of the CON group for low-RFI beef steers (86 vs. 131 mM). Interactions of RBP and RFI were observed ($P \leq 0.05$) for mitochondrial activities of complexes IV, V, and mRNA expressions of some immune genes such as *TLR2*, *TLR3*, and *IL23A*. In conclusion, while RBP supplementation did not alter growth performance, its observed effects on hepatic immune gene expression, mitochondrial protein complexes, BUN, and UN depended on the beef steers' RFI phenotype. Therefore, the RFI status of beef steers should be considered in future studies evaluating the effects of dietary protein supplements.

## Introduction

Protein, an essential and expensive nutrient in animal diets, especially for high-performance cattle, is often a limiting factor [1]. Ruminants mainly derive their amino acids from dietary protein that escapes rumen degradation and from microbial proteins [2]. Unlike other amino acids degraded in the rumen, rumen-bypass amino acids maintain their structural integrity, ensuring that they reach the bloodstream intact [3, 4]. To optimize amino acids utilization in ruminants, introducing essential amino acids like lysine and methionine in forms that resist rumen degradation, but remain available for breakdown and absorption in the small intestine is advantageous [1, 4].

Feeding rumen-bypass protein to cattle has positively impacted weight gain and feed efficiency, especially in rapidly growing young calves [2, 5]. Enhancing the accessibility of certain limiting amino acids such as lysine, methionine, and threonine can lead to improved growth and efficiency [6]. However, the increasing costs of feed protein and the challenges of excessive nitrogen discharge into the environment underscore the need for sustainable protein supplementation in ruminant diets [7]. Although past research has aimed at enhancing amino acids utilization efficiency and/or reducing nitrogen excretion in ruminants by minimizing dietary rumen degradable protein content [8, 9], there is a gap: no study has assessed how factors intrinsic to the animal, like residual feed intake (RFI) status, might influence the response of ruminants to dietary protein supplements.

Numerous studies have consistently spotlighted the association between RFI, a measure of feed and metabolic efficiency, and amino acids metabolism. For instance, Elolimy et al [10], observed a connection between RFI and protein turnover and transport mechanisms in the rumen epithelium of beef cattle. Our recent study by Taiwo et al [11] and those of others [12–14] have further illuminated the link between RFI and amino acid metabolism in beef steers. The results of these studies invariably highlight amino acid metabolism as one of the pathways associated with RFI phenotype in beef steers, implying significant disparities in nitrogen utilization efficiency among beef cattle based on RFI variations. In this context, our study hypothesizes a divergent response to dietary supplementation of RBPs between beef steers with low (or negative RFI) and those with high (or positive) RFI. Thus, the objective of this study was to investigate the effects of RBP supplement on growth performance, plasma, and urinary N (UN) concentration, hepatic mitochondrial protein complexes, and mRNA expression of immune genes in crossbred beef steers with low or high RFI phenotype.

## Materials and methods

All animal care and use procedures were in accordance with the guidelines for the use of Animals in Agriculture Teaching and Research as approved by the West Virginia University Animal Care and Use Committee (2204052569). A total of 108 beef steers were fed a corn silage-based diet Table 1 for a period of 63 days to determine their RFI. Details of animal feeding, including RFI determination followed the procedures reported in our previous study by Taiwo et al [11]. Immediately after RFI determination, a total of 40 crossbred beef steers (BW = 492 ± 36 kg) with divergent negative (low; average = -2.12) or positive (high; average = 2.02) RFI were selected and placed on the same basal diet for a 42-d study to determine the effects of supplemental RBP. The experiment was a 2 × 2 factorial design where the main factors were: 1. RFI classification; low-RFI (LRFI, n = 20) or high-RFI (HRFI, n = 20). Rumen-bypass protein supplement: corn silage-based diet with no rumen-bypass protein supplement (CON, n = 20) or diet supplemented with rumen-bypass protein (RBP, n = 20 fed at 227 g/steer/d), resulting in four treatments: LRFI-CON (n = 10), LRFI-RBP (n = 10), HRFI-CON (n = 10), and HRFI-RBP (n = 10). The rumen-bypass protein supplement (84% CP) is a blend of hydrolyzed feather meal, porcine blood meal, and DL methionine hydroxy analogue (Papillon Agricultural Company, Easton, MD). The corn silage-based diet was fed as a total mixed ration and was formulated to satisfy the metabolic needs of the beef cattle, as reported in our previous study [15]. Based on the initial BW, the steers were stratified by BW into four pens (one treatment group per pen). Each of the pens (size = 10.9 by 4.7 $m^2$) was equipped with two GrowSafe intake nodes (GrowSafe Systems Ltd., Airdrie, Alberta, Canada) to measure

**Table 1. Ingredient and chemical composition of the basal diet.**

| Ingredients (%DM) | % of dietary DM |
|---|---|
| Corn silage | 41.2 |
| Sorghum haylage | 29.4 |
| Mixed grass hay[a] | 10.3 |
| Concentrate supplement[b] | 19.1 |
| Nutrient analysis | |
| DM, % | 52.0 |
| Crude protein, % | 14.1 |
| RUP, % CP | 41.9 |
| RDP, % CP | 58.1 |
| NDF, % | 36.5 |
| NFC, % | 38.0 |
| Fat, % | 4.45 |
| Calcium, % | 0.53 |
| Phosphorus, % | 0.46 |
| Potassium, % | 1.50 |

[a]Contains a mixture of Orchard grass, fescue grass, timothy grass, and red clover

[b]Traditions 50% beef supplement (Southern States Cooperative, Richmond, VA) contained processed grain by-products, plant protein products, ground limestone, urea, salt, cane molasses, potassium sulfate, magnesium sulfate, sodium selenite, vitamin A supplement, calcium carbonate, vegetable oil, manganous oxide, vitamin D3 supplement, vitamin E supplement, zinc oxide, lecithin, phosphoric acid, basic copper chloride, magnesium chloride, propylene glycol, natural and artificial flavors, ferrous sulfate, calcium iodate, and cobalt carbonate; Guaranteed analysis: 50% CP; 5% Ca; 0.55% P; 2% Na; 3.9% salt; 1% K, and 66,000 IU/kg vitamin A

DM, dry matter; CP, crude protein; RUP, rumen undegradable protein; RDP, rumen degradable protein; NDF, neutral detergent fiber; NFC, non-fiber carbohydrates.

individual feed intake. The dietary plan was devised to incorporate 227 grams per day per steer of the RBP. The RBP was mixed in the TMR at a specific proportion, determined by considering the average intake from the previous day for each pen. This intake data from day "x" was utilized to compute the inclusion rate for the subsequent day (x + 1), ensuring the necessary RBP amount for each steer in every pen (averaging 227 g of the specified RBP per head per day). The CON and RBP diets were mixed in separate feed trucks to eliminate the possibility of cross-contamination of diets, and the diets were fed ad libitum (to achieve approximately 10% ort) daily at 0900 h.

## Body weight and intake measurement

Body weights and feed intake (or dry matter intake) of steers were obtained before morning feeding on d 0, 7, 14, 21, 28, 35, and 42. Individual feed intake was measured using the Grow-Safe intake nodes (GrowSafe Systems Ltd., Airdrie, Alberta, Canada). Average daily gain (ADG) was determined by subtracting the initial weight from the final weight on day 42 and then dividing it by the duration of the experiment (42). Samples of TMR were collected daily, weighed and oven dried at 55˚C for 72 h to determine DM content. Subsamples of the dried TMR were composited within treatment, ground using a Wiley mill (Arthur H. Thomas Co., Philadelphia, PA) to pass a 2-mm sieve and sent to a commercial laboratory (Dairy One Forage Laboratory, Ithaca, NY) for analysis for nutritional composition.

## Blood and urine collection

Blood samples (10 mL each) were collected from each steer before the morning feeding via jugular venipuncture on d 0 and 42. Immediately after collection, plasma samples were prepared by centrifugation at $2,500 \times g$ for 20 min at 4˚C and stored at −80˚C until analyzed. Urine samples were collected on d 0 and 42 through manual compression of the urinary bladder, utilizing a mid-stream collection method, performed prior to the morning feeding. These samples were then deposited into sterile 100-mL urine vials (SNL Quality, Delaware, USA). Following collection, the urine samples were promptly placed on ice and within a 30-minute timeframe stored at -80˚C until subsequent analysis. Both plasma and urine samples were analyzed for urea N in duplicates using a microplate reader (Synergy HTX; BioTek Instruments, Winooski, VT) as described by Jung et al [16] and modified by Zawada et al [17]. Plasma samples were deproteinized with an equal volume of 0.6 N $HClO_4$. Urine samples were diluted 100× with distilled water prior to the analysis.

## Liver biopsy

On the last day of the experiment, liver biopsies were obtained from all beef steers while under local anesthesia. The skin was incised, and liver tissue was collected using a 14-gauge biopsy needle (Tru-Core-II Automatic Biopsy Instrument: Angiotech, Lausanne, Switzerland). Approximately 1g of liver tissue samples obtained by one puncture were immediately stored in sterile 2ml tubes containing RNA later solution to stabilize intracellular RNA in tissue samples. These samples were stored at −80˚C until RNA extraction and expression analysis of some immune-related genes.

## Hepatic tissue immune gene expression analysis

Total RNA was then extracted from (∼5mg) of the liver tissue samples collected using RNeasy Micro kit (cat. no. 74004; Qiagen) strictly following the manufacturer's instructions. Samples with >100 ng/μL total RNA were used. The RNA concentration was measured using a

NanoDrop 2000 spectrophotometer with an A260:A280 ratio from 1.8 to 2.0 (Thermo Fisher Scientific, Waltham, MA, USA) and RNA integrity number (> 8.0) was verified using Agilent 2100 bioanalyzer (Agilent Technologies; Santa Clara, CA). After evaluation of RNA purity and final concentration, cDNA was synthesized through reverse transcription (RT) using the RT$^2$ First Strand Kit (cat. no. 330401; Qiagen) following the manufacturer's instructions. Thereafter, mRNA expression of 16 immune-related genes (*CD40*, *LY96*, *NOD2*, *TLR2*, *TLR6*, *TLR3*, *TLR7*, *TLR9*, *IL1R1*, *IL6*, *IL13*, *IL15*, *IL17A*, *IL18*, *IL1A*, and *IL23A*) was analyzed using a Qiagen PCR array, This array also contains five housekeeping genes (β-actin, hypoxanthine phosphoribosyl-transferase 1, glyceraldehyde-3-phosphate dehydrogenase, and tyrosine 3-monooxygenase, TATA box-binding protein), three RT, three positive PCR controls, and one genomic DNA control (S1 Table). Real-time PCR was performed using a QuantStudio 5 Real-Time PCR System (Applied Biosystems, Foster City, CA). The PCR cycling conditions were as follows: 95˚C for 10 min, 40 cycles of denaturation at 95˚C for 15 s, and 60˚C for 1 min.

## Mitochondrial isolation

Liver biopsies (∼ 5 mg) were excised and stored in MAS buffer (a nonionic mannitol, and sucrose-based buffer used in permeabilization assays) at −80˚C until it was subjected to differential centrifugation to isolate mitochondrial, cytosolic, and nuclear fractions. The isolation of subsarcolemmal and interfibrillar mitochondrial subpopulations followed the established method described by Palmer et al [18], with modifications introduced by Baseler et al [19]. The isolated mitochondria were then suspended in KME buffer (100 mM KCl, 50 mM MOPS, and 0.5 mM EGTA, pH 7.4). Protein concentrations were determined using the Bradford method, utilizing bovine serum albumin as a standard [20].

## Mitochondrial electron transport chain (ETC) complex activities

Activities of the mitochondrial electron transport chain complexes (I, II, III, IV, V) were assessed in the liver following previously established protocols [21, 22]. Whole liver tissue was homogenized using the Polytron PowerGen 500 S1 tissue homogenizer (Fisher Scientific, Hampton, NH) in Radioimmunoprecipitation assay buffer (LifeTechnologies, Grand Island, NY). Protein content normalization of the samples was performed using the Bradford assay [20]. The activities of complex I (decyclubiquinone reduction), complex II (succinate dehydrogenase), complex III (cytochrome c reduction), complex IV (reduced cytochrome c oxidation), and complex V (pyruvate kinase and phosphoenolpyruvate and ATP production) were measured using the following equation;

Protein mitochondrial ETC complex activity (nmol/min/mg) = ($\triangle$Absorbance/ min × 1,000)/[(extinction coefficient × sample volume, ml) × (sample protein concentration in mg ml$^{-1}$)];

Where the extinction coefficients in the specified complexes are as follows: for NADH in complexes I and V: 6.2 mM$^{-1}$ cm$^{-1}$, for DCIP in complex II: 19.1 mM$^{-1}$ cm$^{-1}$, and for reduced cytochrome c in complexes III and IV: 18.5 mM$^{-1}$ cm$^{-1}$. The final values were expressed as nanomoles consumed per minute per milligram of protein. This is equivalent to the nanomoles of NADH oxidized per minute per milligram of protein.

## Data and statistical analysis

All data were analyzed using the GLIMMIX procedure of SAS. The statistical model included the fixed effects of RFI phenotype (low-RFI vs. high-RFI), dietary supplementation of rumen-bypass (CON and RBP), interaction between RFI phenotype and dietary supplementation, and the random effect of animal. For the final BW, the initial BW was used as a covariate. For

analysis of mRNA expression, the data were first analyzed using the Qiagen web-based platform, GeneGlobe (https://geneglobe.qiagen.com). The comparative cycle threshold (Ct) method was used for the relative quantification of gene expression [23]. Delta-delta-Ct ($^{\Delta\Delta}$Ct) method with normalization of the raw data using the geometric mean of the five housekeeping genes was used to calculate the differences in mRNA expression of the genes [23]. The stability of the reference genes was confirmed using $^{\Delta}$Ct and NormFinder [24, 25]. The data, depicted as fold-change relative to high-RFI beef steers fed the control diet, were analyzed with the same model described previously. Evidence of statistical significance was considered at $P \leq 0.05$, and tendency was considered at $0.05 < P \leq 0.10$.

## Results

### Growth performance

The results of the growth performance of the beef steers are shown in Table 2. The RFI status, feeding the supplemental RBP or the RFI × RBP interaction did not affect ($P > 0.05$) the final BW or ADG of the beef steers, respectively. However, low-RFI beef steers tended to have lower DMI than high-RFI beef steers ($P = 0.07$).

### Mitochondrial electron transport chain (ETC) complex activities

Table 3 shows the results of the hepatic mitochondrial protein complexes. An interaction between RFI and RBP was observed ($P = 0.01$) for the activity of mitochondrial complex V. Specifically, feeding RBP diet resulted in a reduction in mitochondrial complex V activity in high-RFI beef steers, whereas the activity was greater than that of the CON group in low-RFI beef steers. Additionally, an interaction between RFI and RBP ($P = 0.05$) was observed for the activity of complex IV. In high-RFI beef steers fed the RBP diet, the activity of complex IV was increased compared to the CON group. Conversely, in low-RFI beef steers, the activity of complex IV was lower in the RBP group compared to the CON group.

**Hepatic mRNA expression of selected immune genes.** An interaction between RFI and RBP was observed ($P < 0.05$) for hepatic mRNA expression of *TLR2*, *TLR3*, and *IL23A* Table 4. Feeding RBP diet increased ($P < 0.05$) the mRNA expression of *TLR2* (FC = 1.57 vs 0.98), *TLR3* (FC = 1.49 vs 0.97), and *IL23A* (FC = 1.35 vs 1.07) in high-RFI beef steers but the expression of these genes were not altered in low-RFI beef steers. Regardless of the RFI status, feeding dietary RBP diet increased ($P < 0.05$) the hepatic mRNA expression of *IL13* and *IL1A*.

**Table 2. Effect of the supplementation of a rumen-bypass protein supplement on growth performance of beef steers with divergent residual feed intake phenotype.**

| | High RFI | | Low RFI | | SEM | P-values | | |
|---|---|---|---|---|---|---|---|---|
| | CON | RBP | CON | RBP | | RFI | RBP | RFI*RBP |
| RFI (kg/d) | 2.41 | 1.64 | -2.29 | -1.94 | 0.76 | 0.01 | NA | NA |
| Initial body weight (kg) | 489 | 500 | 490 | 487 | 16.9 | 0.95 | 0.84 | 0.54 |
| Final body weight (kg) | 552 | 563 | 555 | 553 | 2.56 | 0.19 | 0.18 | 0.74 |
| ADG (kg/d) | 1.49 | 1.49 | 1.53 | 1.57 | 0.05 | 0.70 | 0.29 | 0.74 |
| DMI (kg/d) | 13.5 | 13.7 | 12.7 | 13.0 | 0.55 | 0.07 | 0.49 | 0.83 |

CON, control; RBP, a blend of hydrolyzed feather meal, porcine blood meal, DL methionine Hydroxy Analogue, and Ethoxyquin fed at 227 g/steer/d (Papillon, Easton, MD); SEM, standard error of the mean; RFI, residual feed intake; ADG, average daily gain; DMI, dry matter intake; NA, not applicable.

**Table 3. Effect of the supplementation of a rumen-bypass protein supplement on the mitochondrial electron transport chain activities of complexes I–V (nmol/min/mg) of beef steers with divergent residual feed intake phenotype.**

| | High RFI | | Low RFI | | SEM | P-values | | |
| --- | --- | --- | --- | --- | --- | --- | --- | --- |
| | CON | RBP | CON | RBP | | RFI | RBP | RFI*RBP |
| Complex I | 126 | 126 | 151 | 122 | 15.3 | 0.11 | 0.07 | 0.18 |
| Complex II | 73.9 | 75.0 | 70.2 | 68.9 | 6.88 | 0.59 | 0.85 | 0.81 |
| Complex III | 439 | 477 | 408 | 391 | 42.0 | 0.46 | 0.69 | 0.37 |
| Complex IV | 23.9 | 39.2 | 25.4 | 13.1 | 9.63 | 0.88 | 0.22 | 0.05 |
| Complex V | 135 | 94.1 | 76.6 | 120 | 15.8 | 0.01 | 0.01 | 0.01 |

CON, control; RBP, a blend of hydrolyzed feather meal, porcine blood meal, DL-methionine Hydroxy Analogue, and Ethoxyquin fed at 227 g/ steer/d (Papillon, Easton, MD); RFI, residual feed intake.

## Blood and urine urea nitrogen

Supplemental RBP increased ($P = 0.01$) the BUN of the beef steers compared to CON; however, the BUN concentration was lower ($P = 0.01$) in low-RFI compared to high-RFI beef steers Table 5. A tendency for interaction between RFI and RBP ($P = 0.07$) was observed for UN concentration. Specifically, compared to the CON group, feeding the RBP diet increased UN concentration in high-RFI beef steers (124 vs. 209 mM), whereas the concentration was lower than that of the CON group in low-RFI beef steers (86 vs. 131 mM).

## Discussion

This study aimed to determine whether the RFI status of beef cattle would influence their response to dietary RBP supplementation. Our results showed no effects of RFI or RBP

**Table 4. Effect of the supplementation of a rumen-bypass protein supplement on hepatic immune gene expression in beef steers with divergent residual feed intake phenotype.**

| | High RFI | | Low RFI | | SEM | P-values | | |
| --- | --- | --- | --- | --- | --- | --- | --- | --- |
| | CON | RBP | CON | RBP | | RFI | RBP | RFI*RBP |
| CD40 | 1.00 | 1.05 | 1.20 | 1.10 | 0.13 | 0.12 | 0.44 | 0.41 |
| LY96 | 1.00 | 0.98 | 1.09 | 0.98 | 0.14 | 0.52 | 0.43 | 0.63 |
| NOD2 | 1.00 | 0.76 | 1.05 | 1.34 | 0.25 | 0.85 | 0.25 | 0.14 |
| TLR2 | 1.00 | 1.57 | 0.99 | 0.98 | 0.17 | 0.12 | 0.16 | 0.05 |
| TLR6 | 1.00 | 1.17 | 1.28 | 0.99 | 0.18 | 0.11 | 0.11 | 0.17 |
| TLR3 | 1.00 | 1.49 | 0.96 | 0.97 | 0.15 | 0.78 | 0.92 | 0.04 |
| TLR7 | 1.00 | 1.62 | 1.23 | 1.96 | 0.47 | 0.62 | 0.13 | 0.86 |
| TLR9 | 1.00 | 2.77 | 1.43 | 3.60 | 1.43 | 0.76 | 0.14 | 0.84 |
| IL1R1 | 1.00 | 1.07 | 0.97 | 0.87 | 0.14 | 0.81 | 0.49 | 0.41 |
| IL6 | 1.00 | 1.57 | 1.30 | 1.63 | 0.47 | 0.52 | 0.49 | 0.71 |
| IL13 | 1.00 | 3.05 | 1.77 | 3.59 | 0.79 | 0.62 | 0.01 | 0.92 |
| IL15 | 1.00 | 0.87 | 0.99 | 0.84 | 0.09 | 0.95 | 0.11 | 0.89 |
| IL17A | 1.00 | 2.65 | 1.86 | 4.21 | 1.68 | 0.60 | 0.17 | 0.76 |
| IL18 | 1.00 | 1.29 | 1.07 | 1.15 | 0.20 | 0.72 | 0.68 | 0.47 |
| IL1A | 1.00 | 1.21 | 1.22 | 1.73 | 0.24 | 0.36 | 0.04 | 0.36 |
| IL23A | 1.00 | 1.75 | 1.06 | 1.07 | 0.31 | 0.94 | 0.98 | 0.02 |

CON, control; RBP, a blend of hydrolyzed feather meal, porcine blood meal, DL-methionine Hydroxy Analogue, and Ethoxyquin fed at 227 g/ steer/d (Papillon, Easton, MD); RFI, residual feed intake.
Results are depicted as fold-change relative to high-RFI beef steers fed the control diet.

**Table 5. Effect of the supplementation of a rumen-bypass protein supplement on the blood urea nitrogen, and urinary urea nitrogen concentration in beef steers with divergent residual feed intake phenotype.**

|  | High RFI | | Low RFI | | SEM | P-values | | |
|---|---|---|---|---|---|---|---|---|
|  | CON | RBP | CON | RBP |  | RFI | RBP | RFI*RBP |
| Blood urea (mM) | 6.02 | 7.35 | 5.34 | 6.48 | 0.28 | 0.01 | 0.01 | 0.74 |
| Urinary urea (mM) | 124 | 209 | 131 | 86 | 33.8 | 0.26 | 0.34 | 0.07 |

CON, control; RBP, a blend of hydrolyzed feather meal, porcine blood meal, DL-methionine Hydroxy Analogue, and Ethoxyquin fed at 227 g/steer/d (Papillon, Easton, MD); RFI, residual feed intake.

supplementation, or their interaction, on the ADG of the beef steers. However, as expected, low-RFI beef steers tended to have lower DMI than high-RFI beef steers, regardless of RBP supplementation. Previous studies [26, 27] demonstrated that the performance of growing and finishing cattle (dairy or beef) was not significantly influenced by amino acid supplementation. However, it was noted that (dairy or beef) cattle supplemented with amino acids tended to have slightly higher carcass fat content, suggesting a potential improvement in energy availability due to increased fatty acid oxidation [28]. Campbell et al [29] suggested that different amino acid compositions or higher supplementation levels could lead to diverse effects, potentially impacting ruminal fermentation, diet digestibility, and nitrogen flow to the small intestine, ultimately influencing overall performance. It could be inferred that beef steers receiving the control diet in this study were already meeting their amino acid needs. Consequently, greater quantities of the RBP supplement or a longer duration of supplementation may be necessary to observe effects on the performance of the beef steers, as suggested by Hoshford et al [30].

The dietary proteins and their amino acids play a pivotal role in antioxidant mechanisms and serve as substrates for energy provision, contributing significantly to whole-body energy equilibrium, growth, and development [31]. Variations in efficiency within animal populations, not solely linked to pre-digestive factors such as intake or digestibility, suggest the presence of differences in energetic efficiency at the mitochondrial level [32]. Considering that cellular metabolic processes can explain over 50% of the variability in RFI and given that mitochondria primarily produce the majority of cellular energy [33], it has been postulated that a connection exists between RFI and mitochondrial function. Indeed, increased muscle mitochondrial activity was observed in feed-efficient Angus beef steers compared to inefficient ones [34]. Moreover, it was observed that the protein abundance of a mitochondrial component, specifically within Complex I, was significantly higher in low-RFI beef cattle when compared to high-RFI cattle [32]. This observation led to the conclusion that differences in mitochondrial function are associated with variations in metabolic efficiency among animals. Consequently, mitochondrial function could serve as a potential tool for selecting animals based on their efficiency in various metabolic processes [32]. Mitochondrial protein complex IV, also known as cytochrome c oxidase, facilitates the transport of four protons across the membrane, thereby increasing the transmembrane gradient of proton electrochemical potential [35, 36]. This potential difference is subsequently harnessed by ATP synthase for the synthesis of ATP [37]. Consequently, increased activity of Complex IV may lead to more efficient electron transfer, ultimately resulting in an increased generation of ATP and elevated energy production within the mitochondria.

These improvements in mitochondrial functionality can have a range of physiological effects and are often associated with increased cellular energy metabolism. Therefore, the increased mitochondrial Complex IV activity observed in high-RFI beef steers fed the supplemental RBP suggests improved mitochondrial activity. Conversely, the reduced Complex IV

activity in low-RFI beef steers fed supplemental RBP could suggest these beef steers already have optimal mitochondrial energy levels, evidenced by increased Complex V activity. Mitochondrial ATP synthesis is the primary energy reserve for driving intracellular metabolic pathways [38]. Mitochondrial ATP synthase, also known as Complex V in oxidative phosphorylation terms, is the fifth multimeric complex. It primarily synthesizes ATP from ADP in the mitochondrial matrix, leveraging energy from the proton electrochemical gradient [39–41]. Elevated ATP production bolsters cellular energy, aiding numerous metabolic processes, including active transport, biosynthesis, and maintaining ion gradients across cell membranes [42]. Thus, increased Complex V activity in low-RFI beef steers fed the supplemental RBP, compared to CON, may signal an increased ATP production rate. This might partly explain their reduced Complex IV activity since they are presumed to already have optimal mitochondrial energy levels.

In this study, we observed an interaction between RFI status and RBP supplementation on hepatic mRNA expression of immune genes such as *TLR2*, *TLR3*, and *IL-23*. Members of the Toll-like receptor family are known to play a crucial role in pathogen recognition and the activation of innate immunity. *TLR2* is known to play a pivotal role as a primary pattern recognition receptor in the distinct immune response of mammals to mycobacterial antigens, and it also mediates the production of cytokines necessary for effective immunity [43–45]. *TLR3* functions as a modulator of immune response augmentation and operates as an intrinsic detector of tissue necrosis [46]. *TLR3* identifies pathogen-associated molecular patterns (PAMPs) present on infectious agents and facilitates the generation of cytokines essential for the establishment of a robust immune response [47, 48]. *TLR3* also detects double-stranded RNA, a genetic material configuration present in certain viruses like reoviruses [46, 49, 50]. *IL-23* plays a significant role in modulating the inflammatory response to infections and influences various immune processes, such as inflammatory responses, bacterial and antimicrobial defense [51, 52].

Hepatic mRNA expression of these immune genes was upregulated in high-RFI beef steers fed supplemental RBP compared to CON, but these effects were not observed in low-RFI beef steers, indicating that RBP supplementation was effective at improving the ability of high-RFI beef steers to detect and initiate appropriate defense against several infections, especially viral infections. While there is currently no direct link between RFI status and immunocompetence, it's reasonable to assume that low-RFI cattle, which are more feed-efficient and typically in better body condition [53, 54], may have some advantages in terms of overall health and immunocompetence and are believed to be in optimum health status. Prior research has demonstrated that animals with low RFI exhibit an improved capacity to identify pathogens and initiate appropriate responses against these pathogens when compared to those with high RFI [55, 56]. For instance, Taiwo et al [56] compared the hepatic and whole blood mRNA expression of genes associated with innate and adaptive immune response in beef steers with divergent RFI, and the results showed that low-RFI beef steers had greater expression of immune genes associated with pattern recognition receptor activity and T-helper cell differentiation than high-RFI beef steers. This may probably explain why the positive effect of RBP supplementation on these immune genes was observed in high-RFI, but not in low-RFI.

Regardless of dietary RBP supplementation, the low-RFI beef steers exhibited increased mRNA expression of *TLR5* and *CXCL8* genes. *TLR5* interacts with bacterial flagellin, thereby playing a crucial role in defending the host against bacterial pathogens [57, 58], while *CXCL8* plays a crucial role in recruiting and activating immune cells, particularly neutrophils, at sites of infection or tissue damage [59–61], suggesting that low-RFI beef cattle possess a more efficient immune response and host defense against infections. Regardless of RFI status, the hepatic mRNA expression of *IL13* and *IL1A* genes was upregulated in beef steers fed

supplemental RBP compared to CON. *IL-13* and *IL-1* alpha are both cytokines, which are small proteins that play essential roles in the immune system and various physiological processes. *IL-13* has anti-inflammatory effects and is primarily associated with allergic responses and certain immune regulatory functions [62–64], while *IL-1* alpha is a pro-inflammatory cytokine involved in initiating and promoting immune responses and inflammation [65]. Both cytokines are essential for maintaining immune homeostasis. It is well known that amino acids play a crucial role in maintaining immune homeostasis, as they serve as important regulators and signaling molecules within the immune system [31, 66]. In addition, amino acids can influence the development and function of immune cells within the gut-associated lymphoid tissues, maintaining gut immune homeostasis and responding to pathogens [67, 68]. While the primary focus of RBP in animals is often on improving growth and production performance, our results suggest that supplementation of RBP to beef cattle can have beneficial effects on the immune system of beef cattle, regardless of their feed efficiency status.

Urine is composed predominantly of urea and various metabolites of amino acids, making it a valuable sample for evaluating the efficiency of N utilization in ruminants [69, 70]. The majority of urea in the bloodstream is eliminated via urine. When nitrogen intake increases, there is a more noticeable increase in the excretion of total nitrogen through urine compared to fecal nitrogen excretion [71, 72]. Our results revealed that UN concentration of the low-RFI beef steers was lower in those fed supplemental RBP compared to CON. However, in high-RFI beef steers, UN concentration was greater in those fed supplemental RBP compared to CON. This result suggests that low-RFI beef steers may have a higher nitrogen utilization efficiency even with increased nitrogen intake, which may be attributed, in part, to reduced hepatic amino acid catabolism [11, 13]. In our previous study by Taiwo et al [11], hepatic mRNA expression of a gene encoding aminoadipate aminotransferase, an enzyme responsible for lysine degradation, was downregulated in low-RFI, compared to high-RFI beef steers. In agreement with our results, Xie et al [73] revealed that low-RFI dairy cows are more efficient than high-RFI in utilizing metabolizable protein for milk protein and mammary amino acid utilization. In another related study, Rius et al [74] reported that the apparent digestibility of dietary protein was higher in low-RFI, compared to high-RFI dairy cattle, suggesting better N retention and utilization to support production, growth, and other physiological processes.

Blood metabolites such as BUN have been employed as biomarkers for the purpose of monitoring and assessing the health and nutritional status of animals [75], and this could elucidate, to some extent, the biological variability observed in RFI. Regardless of the supplemental RBP, BUN levels in low-RFI beef steers were lower when compared to high-RFI beef steers. This observation aligns with the outcomes reported in previous studies [76–78], which indicated that animals with high-RFI had increased BUN concentrations, relative to those with low-RFI. These findings imply that beef steers with higher RFI values experience a slower rate of protein breakdown compared to more efficient animals with lower RFI values. Bezerra et al [77] stated that low BUN levels in low-RFI steers could be linked to efficient protein deposition in low-RFI animals. Likewise, previous studies suggested that higher BUN levels observed in high-RFI steers could be attributed to several factors, including an increased protein intake in high-RFI animals, a higher rate of body protein degradation, and variations in the supply of amino acids influenced, in part, by variations in the efficiency of microbial protein production in the rumen [79, 80].

## Conclusion

The results of this study revealed that neither supplemental RBP nor the interaction between RFI and RBP exerted observable effects on the growth performance of the beef steers.

However, a notable trend was identified among low-RFI beef steers: supplemental RBP reduced UN levels, compared to those levels in high-RFI counterparts, suggesting that low-RFI beef steers may utilize nitrogen more efficiently, reducing waste while enhancing overall efficiency by potentially increasing nitrogen use efficiency. Remarkably, regardless of dietary RBP supplementation, low-RFI beef steers consistently displayed lower levels of BUN than high-RFI beef steers. Furthermore, the dietary RBP supplementation increased hepatic gene expression related to pathogen recognition and regulation of inflammatory reactions in high-RFI beef steers. This suggests that introducing this feed protein supplement may improve the immune status of these animals. In contrast, low-RFI beef steers probably exhibit optimal immune status, requiring no RBP supplementation to achieve comparable levels of immune functionality. Further studies are warranted to understand the mechanisms underpinning the interactions between RBP supplementation and RFI status.

## Supporting information

**S1 Table. PCR array showing the innate and adaptive immune-related genes.** (XLSX)

## Author Contributions

**Conceptualization:** Modoluwamu Idowu, Francisca Eichie, Ibukun M. Ogunade.

**Data curation:** Modoluwamu Idowu, Andres Pech-Cervantes, Ibukun M. Ogunade.

**Formal analysis:** Modoluwamu Idowu, Ibukun M. Ogunade.

**Funding acquisition:** Ibukun M. Ogunade.

**Investigation:** Modoluwamu Idowu, Godstime Taiwo, Taylor Sidney, Emily Treon, Yarahy Leal, Deborah Ologunagba, Francisca Eichie, Andres Pech-Cervantes.

**Methodology:** Modoluwamu Idowu, Godstime Taiwo.

**Supervision:** Ibukun M. Ogunade.

**Visualization:** Modoluwamu Idowu, Godstime Taiwo, Ibukun M. Ogunade.

**Writing – original draft:** Modoluwamu Idowu.

**Writing – review & editing:** Godstime Taiwo, Taylor Sidney, Emily Treon, Yarahy Leal, Deborah Ologunagba, Francisca Eichie, Andres Pech-Cervantes, Ibukun M. Ogunade.

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
