## [Decision Letter · Decision Letter 0]

21 Dec 2023

PONE-D-23-34051Effects of rumen-bypass protein supplement on growth performance, hepatic mitochondrial protein complexes, and hepatic immune gene expression of beef steers with divergent residual feed intakePLOS ONE

Dear Dr. Ogunade,

Thank you for submitting your manuscript to PLOS ONE. After careful consideration, we feel that it has merit but does not fully meet PLOS ONE’s publication criteria as it currently stands. Therefore, we invite you to submit a revised version of the manuscript that addresses the points raised during the review process.

Your manuscript requires significant revisions. Key points include: standardizing terminology, resolving P-value discrepancies, clarifying the specifics of the rumen-protected amino acid additive, adhering to a consistent format, and providing a clear rationale for your experimental choices. Please address these issues for improved clarity and coherence.

We look forward to receiving your revised manuscript.

Kind regards,

Arda Yildirim, Ph.D.

Academic Editor

PLOS ONE

“This study was funded by West Virginia University Experimental Station in support of U.S. Department of Agriculture hatch multi-state regional project W-3010.”

“This study was funded by West Virginia University Experimental Station in support of U.S. Department of Agriculture hatch multi-state regional project W-3010.”

“This study was funded by West Virginia University Experimental Station in support of U.S. Department of Agriculture hatch multi-state regional project W-3010.”

5. We note that your Data Availability Statement is currently as follows: [All relevant data are within the manuscript and its Supporting Information files.]

6. Please upload a copy of Supporting Information Table which you refer to in your text on page 8.

Additional Editor Comments:

Dear Authors,

Your manuscript "Effects of rumen-bypass protein supplement on growth performance, hepatic mitochondrial protein complexes, and hepatic immune gene expression of beef steers with divergent residual feed intake" has been reviewed. Based on the feedback, substantial revisions are required:

Clarify and be consistent with terminology, specifically use "protein supplement" instead of "supplemental additive".

Address discrepancies in P-values mentioned in the text and tables.

Review and revise the content for accuracy and clarity, especially in lines 216-221 and 233-230.

Include detailed and relevant information on the rumen-protected amino acid additive.

Define abbreviations at their first occurrence.

Ensure a consistent format throughout the manuscript, including decimal places in tables.

Adhere to the “Vancouver” citation style and reevaluate the number of references used.

Include key data such as amino acid concentrations or levels of rumen-degradable and undegradable protein, especially in tables.

Provide a clear rationale for your choice of animal model and feeding practices.

Your attention to these points will greatly enhance the clarity and impact of your research. Thanks.

Reviewers' comments:

Reviewer's Responses to Questions

**Comments to the Author**

1. Is the manuscript technically sound, and do the data support the conclusions?

Reviewer #1: Yes

Reviewer #2: Partly

2. Has the statistical analysis been performed appropriately and rigorously? 

Reviewer #1: Yes

Reviewer #2: Yes

3. Have the authors made all data underlying the findings in their manuscript fully available?

Reviewer #1: Yes

Reviewer #2: No

4. Is the manuscript presented in an intelligible fashion and written in standard English?

Reviewer #1: Yes

Reviewer #2: Yes

5. Review Comments to the Author

Reviewer #1: The article requires minor revision

1. use "protein supplement" instead of "supplemental additive"

2. line 227: "there was a trend (P=0.09), however, in Table 3 we can see P=0.05?????

3. line 216 - 221: Delete it

4; line 233 - 230: Move it to line 216 and correct the P value according to Table 3 (P=0.05 ionstead of 0.09)

5. line 347: Replace the sentence "Urea is........excretion by: The majority of urea in the bloodstream is eliminated via urine

Reviewer #2: The manuscript describes the consequences of supplementation of rumen-bypass protein supplement or a rumen-protected amino acid additive. The research question is interesting, a novel and essential topic, but there are several concerns with the manuscript at this stage. I indicated my special comments in the docx file (CP MANUSCRIPT 10 Reviewed.docx). Unfortunately, these challenges combined would require a significant revision to the manuscript before it reached the standards of PLOS ONE.

6. PLOS authors have the option to publish the peer review history of their article (what does this mean?). If published, this will include your full peer review and any attached files.

Reviewer #1: **Yes: **FLÁVIO A. PORTELA SANTOS

Reviewer #2: No

---

## [Author Response · Author response to Decision Letter 0]

13 Feb 2024

Editor’s comments

Your manuscript "Effects of rumen-bypass protein supplement on growth performance, hepatic mitochondrial protein complexes, and hepatic immune gene expression of beef steers with divergent residual feed intake" has been reviewed. Based on the feedback, substantial revisions are required:

Clarify and be consistent with terminology, specifically use "protein supplement" instead of "supplemental additive".

AU: Supplemental additive has been updated as a protein supplement throughout the manuscript.

Address discrepancies in P-values mentioned in the text and tables.

AU: Thank you for catching that. We’ve updated that, find in line 225.

Review and revise the content for accuracy and clarity, especially in lines 216-221 and 233-230. Include detailed and relevant information on the rumen-protected amino acid additive.

Define abbreviations at their first occurrence.

AU: This has been corrected. Detailed and relevant information of the rumen bypass proteins, which is a commercial product, have been added. Lines 113 – 114.

Ensure a consistent format throughout the manuscript, including decimal places in tables.

Adhere to the “Vancouver” citation style and reevaluate the number of references used.

Include key data such as amino acid concentrations or levels of rumen-degradable and undegradable protein, especially in tables.

AU: Reference format has been updated to the “Vancouver” citation style.

Provide a clear rationale for your choice of animal model and feeding practices.

AU: This has been added based on the information of our previous study that demonstrated that amino acid metabolism is the most significant pathway associated with RFI variation in beef cattle. See lines 89 – 91.

Reviewer #2: The manuscript describes the consequences of supplementation of rumen-bypass protein supplement or a rumen-protected amino acid additive. The research question is interesting, a novel and essential topic, but there are several concerns with the manuscript at this stage. I indicated my special comments in the docx file (CP MANUSCRIPT 10 Reviewed.docx). Unfortunately, these challenges combined would require a significant revision to the manuscript before it reached the standards of PLOS ONE.

Reviewer #1:

1. use "protein supplement" instead of "supplemental additive"

AU: Supplemental additive has been updated as protein supplement throughout the manuscript.

2. line 227: "there was a trend (P=0.09), however, in Table 3 we can see P=0.05?????

AU: Thank you for catching that. We’ve updated that, find in line 231.

3. line 216 - 221: Delete it.

AU: Deleted

4; line 233 - 230: Move it to line 216 and correct the P value according to Table 3 (P=0.05 instead of 0.09)

AU: This has been corrected.

5. line 347: Replace the sentence "Urea is........excretion by: The majority of urea in the bloodstream is eliminated via urine

AU: I’ve updated text to “The majority of urea in the bloodstream is eliminated via urine”. See line 346

---

## [Decision Letter · Decision Letter 1]

21 Feb 2024

Effects of rumen-bypass protein supplement on growth performance, hepatic mitochondrial protein complexes, and hepatic immune gene expression of beef steers with divergent residual feed intake

PONE-D-23-34051R1

Dear Dr. Ogunade,

We’re pleased to inform you that your manuscript has been judged scientifically suitable for publication and will be formally accepted for publication once it meets all outstanding technical requirements.

Kind regards,

Arda Yildirim, Ph.D.

Academic Editor

PLOS ONE

Additional Editor Comments (optional):

Dear Dr. Ibukun Ogunade and Team,

This decision comes after careful consideration of the revisions you have made in response to the reviewers' comments and suggestions. Your manuscript has shown significant improvement from its previous version, demonstrating exciting findings and a sound experimental design. I appreciate your efforts in reviewing and accepting all the comments and suggestions provided by the reviewers, which have undoubtedly enhanced the quality and impact of your research. I would like to thank you for your comprehensive revisions and for addressing the concerns raised during the review process. Your willingness to incorporate feedback and refine your manuscript has been instrumental in reaching this positive outcome. The next steps regarding the publication process, including proofreading, layout, and finalization, will be communicated to you shortly by our editorial team. Please be prepared to respond promptly to any requests or queries they may have to ensure a smooth and timely publication process. Congratulations on this accomplishment. I believe your research will make a valuable contribution to "Ruminant Nutrition" and look forward to its publication in Plos One. Should you have any questions or require further information, please do not hesitate to contact me. Regards, Dr. Arda Yıldırım

Reviewers' comments:

Reviewer's Responses to Questions

**Comments to the Author**

1. If the authors have adequately addressed your comments raised in a previous round of review and you feel that this manuscript is now acceptable for publication, you may indicate that here to bypass the “Comments to the Author” section, enter your conflict of interest statement in the “Confidential to Editor” section, and submit your "Accept" recommendation.

Reviewer #2: All comments have been addressed

2. Is the manuscript technically sound, and do the data support the conclusions?

Reviewer #2: Yes

3. Has the statistical analysis been performed appropriately and rigorously? 

Reviewer #2: Yes

4. Have the authors made all data underlying the findings in their manuscript fully available?

Reviewer #2: Yes

5. Is the manuscript presented in an intelligible fashion and written in standard English?

Reviewer #2: Yes

6. Review Comments to the Author

Reviewer #2: Having a novelty, this manuscript reports exciting findings and a sound experimental design. The last version of the manuscript shows significant improvement from the previous version because the comments requested are now included. I thank the authors for reviewing and accepting all the comments and suggestions except for the number of references.

7. PLOS authors have the option to publish the peer review history of their article (what does this mean?). If published, this will include your full peer review and any attached files.

Reviewer #2: No

---

## [Editor Report · Acceptance letter]

24 Jun 2024

PONE-D-23-34051R1 

PLOS ONE

Dear Dr. Ogunade, 

I'm pleased to inform you that your manuscript has been deemed suitable for publication in PLOS ONE. Congratulations! Your manuscript is now being handed over to our production team.

Kind regards, 

on behalf of

Prof. Dr. Arda Yildirim 

Academic Editor

PLOS ONE